# Comparison of five different disseminated intravascular coagulation criteria in predicting mortality in patients with sepsis

**Amara Zafar**[ID], **Filza Naeem**[ID], **Muhammad Zain Khalid**[ID], **Safia Awan, Muhammad Mehmood Riaz, Saad Bin Zafar Mahmood**[ID]*

Department of Medicine, Aga Khan University Hospital, Karachi, Pakistan

* saadbin.zafar@aku.edu

**Data Availability Statement:** All relevant data are within the paper and its Supporting Information files.

## Abstract

### Objective

Even though patients with sepsis and DIC have a higher mortality rate compared to those without DIC, screening for DIC is not currently part of sepsis management protocols. This may be due to a lack of literature on the frequency of DIC occurrence in sepsis patients, as well as the absence of evidence on the optimal DIC criteria to use for identifying DIC and predicting mortality among the five criteria available. To address this gap, this study investigates the predictive value of five different criteria for diagnosing DIC and its relationship to patient outcomes in our population of sepsis patients.

### Methods

In the Medicine department of Aga Khan University Hospital, a retrospective observational study was conducted, enrolling all adult patients with International Classification of Diseases, 9th Revision, Clinical Modification (ICD-9-CM) coding of sepsis and clinical suspicion of DIC between January 2018 and December 2020. To diagnose DIC, five different criteria were utilized, namely the International Society of Thrombosis and Hemostasis (ISTH), the Korean Society on Thrombosis and Hemostasis (KSTH), the Japanese Association for Acute Medicine (JAAM), the revised-JAAM (RJAAM), and the Japanese Ministry of Health and Welfare (JMHW). The study analyzed the sensitivity, specificity, negative predictive value, positive predictive value, and accuracy of these five criteria, as well as the overall prediction of mortality.

### Results

Of 222 septic patients included in this study with clinical suspicion of DIC, 94.6% of patient had DIC according to KSTH criteria, followed by JAAM (69.4%), ISTH (64.0%), JMHW (53.2%) and lastly R-JAAM (48.6%). KSTH had sensitivity of 95.4% in diagnosing DIC and predicting mortality with a positive predictive value of 70% but specificity of 7.3% only. JAAM had sensitivity of 75.9%, positive predictive value of 75.9% with a specificity of

**Funding:** The authors received no specific funding for this work.

**Competing interests:** The authors have declared that no competing interests exist.

45.5%. ISTH had sensitivity of 69.4%, positive predictive value 75.3% and specificity of 48.5%.

## Conclusion

DIC can impose a significant burden on septic patients and its presence can lead to higher mortality rates. Early detection through screening for DIC in septic patients can potentially reduce mortality. However, it is necessary to identify the most appropriate diagnostic criteria for each population, as various criteria have demonstrated different performance in different populations. Establishing a gold standard for each population can aid in accurate diagnosis of DIC.

## Introduction

Sepsis is a critical medical condition that may lead to several complications, one of which is disseminated intravascular coagulation (DIC) [1]. The International Society on Thrombosis and Hemostasis (ISTH) provided a definition for DIC in 2001, stating that it is an acquired syndrome that involves the intravascular activation of coagulation and a loss of localization due to various causes [2]. DIC is a disorder that can be life-threatening and is associated with poor prognosis in sepsis patients, those with associated DIC have a higher mortality rate compared to patients without DIC. In a study of ICU patients with severe sepsis, the prevalence of DIC was 50.9% and the overall mortality rate was 21.5%. The mortality rate was found to be 17.5% for non-DIC patients and 24.8% for those with DIC complications [3]. Diagnosis of DIC is of crucial importance for ICU and critically ill patients; however, no single laboratory tests for accurate diagnosis of DIC is currently available [4].

The diagnosis of DIC is a challenging task due to the variety of clinical presentations and laboratory abnormalities. Several DIC criteria have been proposed and are currently used in clinical practice and research studies. To date, there are five different diagnostic criteria for DIC available, namely; ISTH (International society of thrombosis and hemostasis), KSTH (Korean society on thrombosis and hemostasis), JAAM (Japanese association for Acute medicine) RJAAM (revised-JAAM) and JMHW(Japanese ministry of health and welfare). In 2001, ISTH published the first international diagnostic criteria for overt DIC based on the modification of JMHW criteria [5]. JAAM criteria was published in 2005 and was revised by Gando et al. in 2006 and revealed that the revised JAAM (rJAAM) score had a better prognostic outcome than the JAAM in ICU patients [6]. In a study done on ICU patients in France, no significant difference between JAAM and ISTH criteria was observed; among 582 patients, 32.1% were diagnosed with DIC according to ISTH-DIC score, and 34.4% according to JAAM-DIC score [7].

A prospective study done in Korea which included 100 patients with severe sepsis/septic shock, compared the five diagnostic criteria of DIC for accurate diagnosis and mortality predictor as an outcome parameter. It concluded that the KSTH and JMHW criteria revealed greater statistical significance (P = 0.007, and 0.479, respectively) when applied on Day 1 and proved to result in overall ICU mortality [4]. This was the only study we can find in literature to have compared the five different DIC criteria together.

DIC screening is not a part of sepsis management yet. This could be attributed to the scarcity in literature about the frequency of occurrence of DIC in patients with sepsis. In a study in Japan, it was observed that DIC screening in ICU patients with sepsis was associated with a

reduction in mortality [8]. In patients with sepsis induced DIC, favorable outcome of anticoagulant therapy on the mortality rate have been reported in literature [9].

To the best of our knowledge, no studies have been done in Pakistan which compare the different criteria of DIC with each other in terms of diagnostic accuracy and relationship with mortality. Therefore, the major objective of our study was to compare the performance of five different DIC criteria in patients with sepsis in our population in accurately diagnosing DIC and predicting outcomes in terms of mortality.

## Material and methods

### Study design/data source and collection

This was a retrospective study conducted in the Medicine Department of Aga Khan University Hospital (AKUH). Aga Khan University Hospital is one of the largest academic tertiary care centers in South Asia. It is a 300 bed facility with a state of the art emergency department and rooms of different acuities; low-monitoring ward beds with a nurse ratio of approximately 5:1, 70 high dependency units with 24 hour cardiac monitoring, non-invasive mechanical ventilation facilities and nurse ratio of 5:2, and 15 intensive care units (ICU) with mechanical ventilator facilities and nurse ratio of 1:1. In addition, the hospital has a rapid response team (ICU nurse and doctor) which is the first responder to hospital areas in case of a medical emergency. The hospital also has a robust transport mechanism for patients who need to be shifted to ICU. Such patients are accompanied by an Advanced cardiac life support (ACLS) certified nurse and doctor along with the RRT nurse. Sepsis was defined as life-threatening organ dysfunction caused by a dysregulated host response to infection [10].

The study included all adult (> 18 years) patients with International Classification of Diseases, 9th Revision, Clinical Modification (ICD-9-CM) coding of sepsis based on the Dombrovskiy definition (Infection codes: 003.1, 020.2, 022.3, 036.2, 036.3, 038.0–038.4, 038.8, 038.9, 054.5, 098.89, 112.5, 785.52, 995.91, 995.92 AND Organ dysfunction codes: 286.6, 286.9, 287.5, 293.0, 297.4, 348.1, 348.3, 427.5, 458.0, 458.8, 458.9, 518.81, 518.82, 570, 572.2, 573.4, 584, 780.01, 785.5, 786.09, 799.1, 796.3) [10] and Modified Martin Criteria (Infection codes: 038.0–038.4, 038.8, 038.9, 003.1, 020.2, 036.2, 036.3, 054.5, 098.89, 112.5, 112.81, 117.9, 790.7, 995.91 AND organ dysfunction codes: 518.81, 518.82, 518.84, 518.85, 786.09, 799.1, 785.5 with all sub codes, 458, 796.3, 584 with all sub codes, 580, 570, 572.2, 573.3, 286.6, 286.9, 287.3–5, 293, 348.1, 348.3, 357.82, 780.01, 780.09, 276.2) [11] from January 2018 to December 2020 were reviewed. Patients in whom the routine DIC laboratory tests (platelet, D-dimer, fibrin/ fibrin degradation product (FDP), prothrombin time (PT), activated partial thromboplastin time (aPTT), and fibrinogen) were done were included in this study.

Information was collected from the study center's administrative database, which is managed by the hospital's Health Information Management System (HIMS) Department. A total of 1999 medical records from the period of 2018 to 2020 were thoroughly reviewed. These records had been coded with ICD-9-CM for sepsis. After applying the inclusion criteria, 222 medical records were found to be eligible and were included in the study. Non-probability consecutive sampling was employed. Medical record files were systematically reviewed and patients meeting inclusion criteria were included. Scores were applied as per the criteria and outcome was noted from the records by the same data collector.

The questionnaire recorded the age and gender of the participants. Systemic Inflammatory Response Syndrome (SIRS) was computed based on the temperature, respiratory rate, heart rate, and the white blood cell count reported when the patients DIC workup was done. The data collection team also took note of the need for intubation and dialysis during the patients' hospital stay, as well as whether they developed septic shock at any point. The study utilized

the initial vital signs recorded in the emergency department and the first lactate measurement during the hospital stay to compare survivors and non-survivors. The SOFA score was calculated using the first recorded PaO2/FiO2, platelet count, bilirubin, MAP (hypotension), Glasgow Coma Score (GCS), and creatinine (or urine output). For coagulation marker comparison, only patients who had their platelet count, PT, aPTT, fibrinogen, d-dimer, and FDP measured were included.

All the elements used to assess DIC criteria for each patient originated from a single day. On the initial day, 44.3% underwent DIC evaluation based on clinical judgment, while on the second day, the assessment was conducted for 33.3%. On the third day, 13.1% underwent evaluation, followed by 4.2% on the fourth day, and 3.4% on the fifth day. The remaining four patients underwent DIC evaluation on days 6, 8, 10, and 11, respectively. Hence, these investigations were carried out on the day when the clinician suspected the patient might be experiencing DIC. These values were not the worst reported values during the admission but were done on a single day. Each patient included underwent evaluation using all five DIC diagnostic criteria: JAAM, R-JAAM, JMHW, ISTH, and KSTH (Table 1, Fig 1).

### Eligibility criteria

Patients with hematologic diseases, underlying bleeding disorders, on medications such as anticoagulation or chemotherapy, Child Pugh grade C liver cirrhosis were excluded from this study.

### Patient and public involvement statement

This study was conducted retrospectively by examining medical records and electronic data. The research did not involve live interviews or direct interaction with patients. Patient confidentiality and anonymity were maintained, and no identifying information that could be used to track participants was included. The study questionnaire was labelled with a serial number. The ethical review committee (ERC) of Aga Khan University Hospital (AKUH), Karachi, Pakistan approved the study as an exemption (IRB reference number: 2021-6280-19084).

### Statistical analysis

All the patients were divided into two groups, survivors and non survivors. Clinical and laboratory parameters were compared between the two groups. The sensitivity, specificity, positive predictive value, negative predictive value, and accuracy of five different DIC diagnostic criteria in terms of overall prediction of mortality were analyzed using the Statistical Package for Social Science (SPSS) version 23. Results were presented as mean ± standard deviation or median with interquartile range (IQR) for continuous variables and frequency (percentages) for categorical variables. Analytical analysis was done according to the study objectives. For comparative analysis, Chi-square, or Fisher's Exact for categorical variables and Mann–Whitney U, or independent sample t-test wherever applicable. All p-values were two-sided and considered as statistically significant if $< 0.05$.

## Results

### Baseline clinical and laboratory characteristics between the survivors and non-survivors of sepsis

Out of 222 patients enrolled in this study, 68 (30.6%) patients survived and 154 (69.4%) did not survive. There was no statistically significant difference observed in mortality in age and gender. The mean age of survivors was 55.1 ± 19.1 years and for non-survivors it was

**Table 1. Summary of five different DIC diagnostic criteria applied in the present study [4].**

| Parameters | Score | KSTH | ISTH | JAAM | JMHW |
|---|---|---|---|---|---|
| **Platelets, × 10⁹/L** | 0 | > 100 | > 100 | ≥ 120 | > 120 |
| | 1 | ≤ 100 | ≤ 100 | ≥80 and < 120 or >30%↓ (≤ 24 hrs) | 80–120 |
| | 2 | | ≤ 50 | | 50–80 |
| | 3 | | | < 80 or > 50%↓ (≤ 24 hrs) | < 50 |
| **PT, sec** | 0 | < 3 | < 3 | < 1.2 (PT ratio) | < 1.25 |
| | 1 | ≥ 3 | ≥ 3 and < 6 | ≥ 1.2 | 1.25–1.67 |
| | 2 | | ≥ 6 | | ≥ 1.67 |
| **aPTT, sec** | 0 | < 5 | | | |
| | 1 | ≥ 5 | | | |
| **Fibrin related marker, µg/mL** | 0 | No increase* | No increase* | No increase‖ | < 10 |
| | 1 | Increase† | | Moderate increase¶ | 10–20 |
| | 2 | | Moderate increase‡ | | 20–40 |
| | 3 | | Marked increase§ | Marked increase** | ≥ 40 |
| **Fibrinogen, g/L** | 0 | > 1.5 | > 1.0 | > 3.5 | > 1.5 |
| | 1 | ≤ 1.5 | ≤ 1.0 | ≤ 3.5 | 1.0–1.5 |
| | 2 | | | | ≤ 1.0 |
| **SIRS score** | 0 | | | 0–2 | |
| | 1 | | | ≥ 3 | |
| **Underlying disease** | 1 | | | | Present |
| **Bleeding** | 1 | | | | Present |
| **Organ failure** | 1 | | | | present |
| **Total** | | DIC ≥ 3 | DIC ≥ 5 | DIC ≥ 5 | DIC ≥ 7 |

R-JAAM criteria have same score system with JAAM except fibrinogen score.

aPTT = activated partial thromboplastin time, DIC = disseminated intravascular coagulation, FDP = fibrin/fibrinogen degradation product, ISTH = International Society on Thrombosis and Haemostasis, JAAM = Japanese Association for Acute Medicine, JMHW = Japanese Ministry of Health and Welfare, KSTH = Korean Society on Thrombosis and Hemostasis, PT = prothrombin time, R-JAAM = Revised JAAM, SIRS = systemic inflammatory response syndrome.

*D-dimer < 1.0

†D-dimer ≥ 1.0

‡1.0 ≤ D-dimer < 5.0

§D-dimer ≥ 5.0

‖FDP < 10

¶10 ≤ FDP < 25

**FDP ≥ 25.

56.9 ± 18.7 years. 89 patients (40.1%) were intubated and experienced an Intensive Care Unit stay.

Patients who required the need of dialysis had higher mortality rates (p-value 0.004). Patients who were intubated, developed septic shock, had higher SOFA and SIRS scores and higher lactate levels were seen to have significantly higher mortality rates (p-value <0.001). The median SOFA score for non-survivors was 13 [11.7–15.0] and for survivors it was 8 [5.0–10.0]. Lactate was higher on initial presentation among non-survivors (median of 5 mM/L [2.0–13.0]) than survivors (median of 2 mM/L [1.0–2.9).

Among infections, genitourinary (GU) infections were significantly associated with mortality (p = 0.030), out of 78 patients with GU infections, 47 (60.3%) patients did not survive and 31 (39.7%) survived. The presence of fungus was found in 50 patients (22.5%) and its presence was significantly associated with mortality, 44 (88%) patients out of 50 did not survive, while

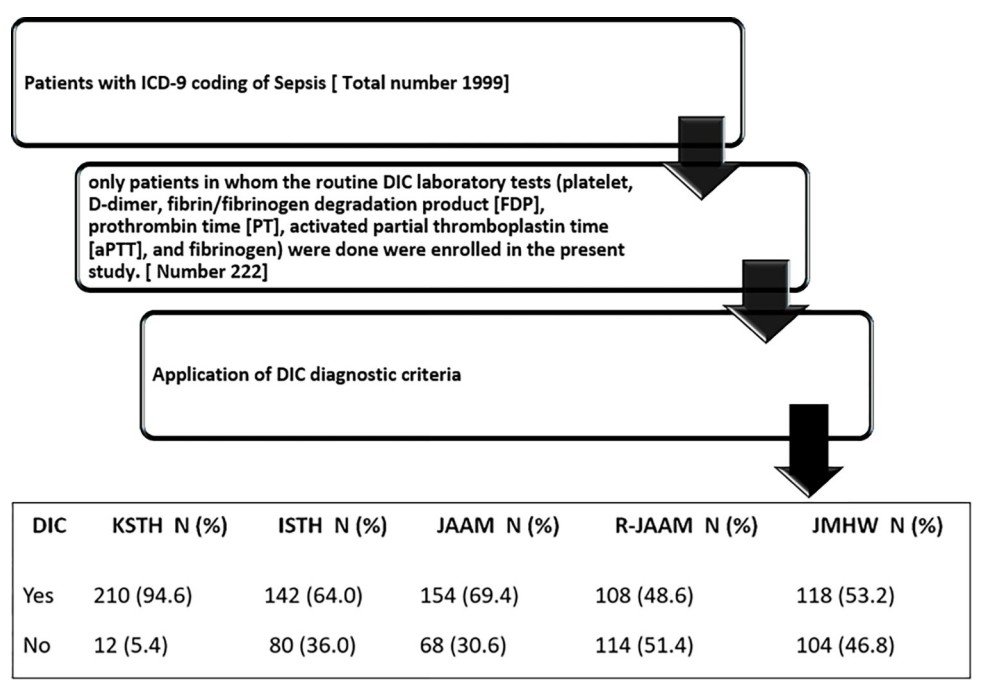

**Fig 1. Selection of sample of the study and application of DIC criteria.**

only 6 (12%) patients survived (p = 0.001). Blood culture was positive in 92 (41.1%) patients, out of which 70 (76.1%) did not survive and 22 (23.9%) survived. There was no significant difference observed in the presence of bacteraemia between survivors and non-survivors (p = 0.068). Out of the 222 patients, 3 (1.4%) had Tuberculosis, 1 (0.5%) had meningitis, 90 (40.5%) had respiratory tract infection (sputum culture positive), 25 (11.3%) had gastrointestinal infection, 78 (35.1%) had GU infection, and 12 (5.4%) had bone and skin infections. Out of 90 patients with respiratory tract infection, 24 (27%) survived and 66 (73%) did not survive (p = 0.290).

When the vital signs of the patients recorded on arrival to the emergency were compared, it was seen that patients who had higher respiratory rate (p<0.001), higher heart rate (p = 0.002) and lower mean arterial pressures (p<0.001) had significantly higher mortality rates.

When coagulation profile was compared, finding of low platelets was significantly associated with mortality (p value = 0.02). High PT (p value = 0.001) and aPTT (p value = 0.004), low fibrinogen (p value = 0.05), and higher FDP (p value = 0.01) were all found to be significantly associated with mortality. D-dimer was the only coagulation marker not found to be significantly associated with mortality (p value = 0.06) (Table 2).

### Comparing DIC diagnosis and prediction of mortality rates using five distinct diagnostic criteria

On application of the five DIC criteria on our study sample, 210 patients (94.6%) had DIC according to KSTH criteria, followed by 154 patients (69.4%) by JAAM, 142 patients (64%) by ISTH, 118 patients (53.2%) by JMHW and 108 patients (48.6%) by R-JAAM. All groups diagnosed with DIC according to different criteria had a high mortality rate. Specifically, the mortality rate was 70% for KSTH, 75.4% for ISTH, 76% for JAAM, 78.7% for R-JAAM, and 78% for JMHW (Table 3).

**Table 2. Comparison of baseline clinical and laboratory characteristics between survivors and non-survivors.**

| Variables | Survivors (n = 68) | Non-survivors (n = 154) | P value |
|---|---|---|---|
| Age, yr | 55.1 ± 19.1 | 56.9 ± 18.7 | 0.52 |
| Sex (M:F ratio) | 32:36 | 85:69 | 0.26 |
| Diabetes Mellitus | 22(32.4) | 73(47.4) | 0.03 |
| Hypertension | 33(48.5) | 73(47.4) | 0.87 |
| **Clinical outcomes** | | | |
| Dialysis, No. (%) | 13(19.1) | 60(39) | **0.004** |
| Ventilator use, No. (%) | 15(22.1) | 74(48.1) | **<0.001** |
| Bacteremia, No. (%) | 22(32.4) | 70(45.5) | 0.06 |
| Septic shock, No. (%) | 55(80.9) | 153(99.4) | **<0.001** |
| SOFA score | 8.0(5.0–10.0) | 13.0(11.7–15.0) | **<0.001** |
| SIRS score | 3.0-(2.0–3.0) | 3.0(3.0–4.0) | **<0.001** |
| Lactate, mM/L | 2.0(1.0–2.9) | 5.0(2.0–13.2) | **<0.001** |
| **Initial vital sign** | | | |
| Respiratory rate/min | 26.0(20.0–26.0) | 28.0(26.0–30.0) | **<0.001** |
| Heart beats/min | 110.0(94.0–115.0) | 113.0(104.7–123.0) | **0.002** |
| Temperature, ˚C | 36.1(36.0–37.0) | 37.0(36.0–38.0) | 0.07 |
| MAP | 65.5(56.0–72.7) | 50.0(45.0–62.2) | **<0.001** |
| **Coagulation markers** | | | |
| Platelet, × $10^9$/L | 82.0(35.7–164.5) | 55.0(31.7–93.2) | **0.02** |
| PT, sec | 16.2(14.0–19.5) | 20.3(14.8–27.3) | **0.001** |
| aPTT, sec | 35.0(27.2–46.1) | 40.7(30.0–62.7) | **0.004** |
| D-dimer, µg/mL | 6.0(2.4–14.3) | 7.7(3.7–19.6) | 0.06 |
| FDP, µg/mL | 5.0(5.0–16.2) | 5.0(5.0–20.0) | **0.01** |
| Fibrinogen, g/L | 258.0(168.2–454.5) | 216.0(131.7–352.7) | **0.05** |

P values were obtained from the Pearson's $\chi^2$ test (for dichotomous variables) and the t-test or Mann-Whitney U test (for continuous variables) and the results were expressed as the mean ± sd or median and interquartile range (IQR, 25th–75th percentile).

aPTT = activated partial thromboplastin, F = female, FDP = fibrin degradation product, M = male, PT = prothrombin time, SOFA = sequential organ failure assessment.

## Performance evaluation of five different DIC diagnostic criteria

In terms of predicting mortality in patients with sepsis, KSTH demonstrated a sensitivity of 95.4% but a specificity of only 7.3%. JAAM had a sensitivity of 75.9%, a specificity of 45.5%, a

**Table 3. Comparison of outcome between different DIC criteria.**

| DIC Criteria- N | | Discharged | Expired |
|---|---|---|---|
| | | N (%) | N (%) |
| KSTH | 210 | 63 (30.0) | 147 (70.0) |
| ISTH | 142 | 35 (24.6) | 107 (75.4) |
| JAAM | 154 | 37 (24.0) | 117 (76.0) |
| R-JAAM | 108 | 23 (21.3) | 85 (78.7) |
| JMHW | 118 | 26 (22.0) | 92 (78.0) |

DIC = disseminated intravascular coagulation, ISTH = International Society on Thrombosis and Haemostasis, JAAM = Japanese Association for Acute Medicine, R-JAAM = Revised JAAM, JMHW = Japanese Ministry of Health and Welfare, KSTH = Korean Society on Thrombosis and Hemostasis.

**Table 4. Performance of five different DIC diagnostic criteria in the prediction of overall mortality.**

| Diagnostic criteria | Sensitivity, % (No.) | Specificity, % (No.) | PPV, % (No.) | NPV, % (No.) | Accuracy, % (No.) |
|---|---|---|---|---|---|
| KSTH criteria | 95.4 | 7.3 | 70.0 | 41.6 | 68.4 |
| ISTH criteria | 69.4 | 48.5 | 75.3 | 41.2 | 63.0 |
| JAAM criteria | 75.9 | 45.5 | 75.9 | 45.5 | 66.6 |
| R-JAAM criteria | 55.1 | 66.1 | 78.7 | 39.4 | 58.5 |
| JMHW criteria | 59.7 | 617 | 77.9 | 40.3 | 60.3 |

ISTH = International Society on Thrombosis and Haemostasis, JAAM = Japanese Association for Acute Medicine, NPV = negative predictive value, PPV = positive predictive value, R-JAAM = revised JAAM, JMHW = Japanese Ministry of Health and Welfare, KSTH = Korean Society on Thrombosis and Hemostasis.

positive predictive value of 75.9%, and an accuracy of 66.6%. ISTH had a sensitivity of 69.4%, a specificity of 48.5%, a positive predictive value of 75.3%, and an accuracy of 63.0%. R-JAAM had the highest specificity (66.1%) and positive predictive value (78.7%) among all five criteria, but its sensitivity (55.1%) and accuracy (58.5%) were low (Table 4).

## Discussion

Sepsis is a life-threatening condition that arises when the body's response to an infection injures its tissues and organs. Mortality in sepsis remains high, and accurate and reliable prognostic tools are essential to identify patients who are at risk of dying. One such tool is the use of different DIC criteria, which are commonly used to assess the severity of coagulopathy in patients with sepsis. Various populations have shown varying results in the diagnosis of DIC when utilizing different criteria.

The purpose of our study was to assess the effectiveness of the five different DIC screening criteria in accurately identifying patients with DIC, as well as to compare the outcomes of these patients. By doing so, we aimed to determine which criteria was the most sensitive in our population. Our study findings indicate that among the evaluated models, KSTH demonstrated superior sensitivity and accuracy, while R-JAAM exhibited the highest specificity and positive predictive value for mortality prediction in sepsis patients as compared to others. Additionally, our study also revealed associated mortality rates, providing important insight into the clinical implications of DIC diagnosis.

As far as we are aware, only one published study has compared the performance of the five DIC criteria in predicting mortality among patients with severe sepsis or septic shock who were admitted to the ICU. The results of this study suggested that both the KSTH and JMHW criteria performed better than the ISTH, JAAM, and R-JAAM criteria in predicting overall ICU and 28-day mortality [4]. Overall, comparison between two studies would be difficult as one was prospective while the other was retrospective. Moreover, it is important to note that each population has different racial characteristics and hence different thrombolytic mechanisms [12].

Our study found that 64.0%, 69.4% and 94.6% of patients were diagnosed with DIC using the ISTH, JAAM and KSTH scoring systems, respectively. Our analysis suggests that the inclusion of SIRS criteria and differences in platelet count and fibrinogen scores may account for the variations in diagnostic outcomes between the criteria. These findings are consistent with a study conducted in a tertiary hospital in France, which reported DIC diagnosis rates of 32.1% using ISTH and 34.4% using JAAM among ICU patients [7].

Another study comparing JAAM and ISTH criteria found that the JAAM DIC scoring system accurately diagnosed all patients with ISTH overt DIC at the time of severe sepsis diagnosis, while the ISTH overt DIC system missed many patients and was unable to detect non-

overt DIC. This indicates that the JAAM system is more effective in selecting DIC patients with a poor prognosis and those requiring treatment. The JAAM system showed high sensitivity and moderate specificity using the ISTH overt DIC criteria as a benchmark. The study suggests that the ISTH overt DIC criteria may be too strict in selecting patients at risk of death and may require the assistance of a non-overt DIC scoring system in a critical care setting [13]. Similarly, another study demonstrated that the JAAM criteria diagnosed all patients identified as having DIC using the KSTH, JMHW, and ISTH criteria as well [4].

A prospective study by the Japanese Society of Thrombosis and Hemostasis (JSTH) assessed three different sets of DIC diagnostic criteria. The JAAM acute phase DIC criteria demonstrated the highest sensitivity, detecting twice as many cases of DIC compared to the other two criteria sets. The JMHW and ISTH overt-DIC criteria followed in sensitivity. Mortality prediction was used as an evaluation criterion since there is no gold standard for diagnosing DIC. The JAAM criteria exhibited the highest sensitivity (80.0%) for predicting mortality but had low specificity (33.2%) [14]. Our study also revealed that JAAM exhibited superior sensitivity (75.9%) in predicting mortality compared to ISTH (69.4%) and JMHW (59.7%). Our study identified low specificity for all three criteria, including JAAM (45.5%), ISTH (48.5%), and JMHW (61.7%). R-JAAM exhibited the highest specificity (66.1%) in our study, while KSTH demonstrated the lowest specificity (7.3%).

The KSTH DIC criteria offer the advantage of being concise and not requiring a weighted score, which makes them a relatively simple option for calculating for the presence of DIC. However, the KSTH criteria are not yet accepted as an international diagnostic method due to a lack of clinical application data to support their use. Several studies have compared the KSTH criteria with other DIC criteria. One study found that the KSTH criteria were more sensitive predictors of mortality than the ISTH criteria, with a concordance rate of 84.7% and a K-coefficient of 0.6 [15]. Another study demonstrated a good level of agreement between the KSTH criteria and ISTH criteria, suggesting that the KSTH criteria may be a useful diagnostic tool for DIC [4]. In our study KSTH had the highest sensitivity (95.4%) but it had the lowest specificity (7.3%) in predicting mortality. Our study included patients who had clinical suspicion of DIC. This reinforces the relevance of utilizing KSTH criteria for our population, as 94.6% of the study sample satisfied the KSTH criteria. This high statistical sensitivity would make KSTH a valuable tool for screening of septic patients having DIC and thus predicting their prognosis.

The establishment of the JMHW criteria marked the first set of criteria for diagnosing DIC. The subsequent global adoption of ISTH overt-DIC criteria as the diagnostic criteria of choice has led to it becoming the widely accepted standard [16, 17].

Although our study provided valuable insights into the diagnostic criteria of DIC in septic patients, it is important to acknowledge its limitations. Firstly, being a single-center study with a limited sample size, the results may not be generalizable to the wider population. Moreover, ICD-9-CM coding for sepsis has been found to be categorized differently across literature and needs to be looked thoroughly between different institutions. Secondly, due to the retrospective nature of the study, only patients who had routine laboratory DIC tests were included, potentially leading to the exclusion of some patients with DIC. The selection of patients who underwent coagulation testing relied on the clinical judgment of physicians, representing a notable constraint in our study. Lastly, our study used hospital course mortality as the outcome measure, without a standardized follow-up period, which may affect the accuracy of our findings. Moreover, our study also did not include the treatment regimens for these patients due to the variability and since there is yet to be an international consensus on whether DIC should be a therapeutic target with anticoagulant therapy [12]. These limitations highlight the need for future studies to further investigate the diagnostic criteria of DIC in septic patients in larger more diverse populations with standardized outcome measures.

The sensitivity and specificity of the various DIC criteria for predicting mortality in sepsis patients differ from one another. While some studies indicate that specific criteria are more precise than others, additional research is necessary to identify the most effective DIC criteria for use in clinical settings. It is essential to validate the optimal cut-off values for laboratory parameters and determine the most precise and dependable DIC criteria for diagnosing and managing patients with this complex medical condition.

## Conclusion

This groundbreaking cohort study is the first of its kind in our country, providing valuable insight into the comparison of the five different diagnostic criteria's of DIC. Our findings serve as a steppingstone towards future prospective studies, ultimately leading to the identification of a gold standard diagnostic criteria best suited for our population. Our study highlights the critical importance of DIC screening as a part of sepsis management, given the high mortality rate observed in patients with sepsis induced DIC. Our study findings indicate that among the five evaluated models of DIC, KSTH demonstrated superior sensitivity and accuracy, while R-JAAM exhibited the highest specificity and positive predictive value for mortality prediction in sepsis patients as compared to others. Our study results highlight the need for large-scale studies to validate these findings before making it a part of local guidelines. The significant burden of DIC in septic patients emphasizes the need for early DIC screening as part of sepsis management, with the potential to reduce mortality rates—an avenue that warrants further exploration.

## Supporting information

**S1 Data.**
(SAV)

## Author Contributions

**Conceptualization:** Amara Zafar, Saad Bin Zafar Mahmood.

**Data curation:** Amara Zafar, Filza Naeem, Muhammad Zain Khalid, Safia Awan, Saad Bin Zafar Mahmood.

**Formal analysis:** Amara Zafar, Safia Awan, Saad Bin Zafar Mahmood.

**Investigation:** Amara Zafar, Safia Awan, Saad Bin Zafar Mahmood.

**Methodology:** Amara Zafar, Safia Awan, Muhammad Mehmood Riaz, Saad Bin Zafar Mahmood.

**Project administration:** Amara Zafar, Filza Naeem, Muhammad Zain Khalid, Muhammad Mehmood Riaz, Saad Bin Zafar Mahmood.

**Supervision:** Amara Zafar, Muhammad Mehmood Riaz, Saad Bin Zafar Mahmood.

**Validation:** Amara Zafar, Muhammad Mehmood Riaz, Saad Bin Zafar Mahmood.

**Visualization:** Amara Zafar, Saad Bin Zafar Mahmood.

**Writing – original draft:** Amara Zafar, Filza Naeem, Muhammad Zain Khalid.

**Writing – review & editing:** Amara Zafar, Muhammad Mehmood Riaz, Saad Bin Zafar Mahmood.

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
