## [Decision Letter · Decision Letter 0]

28 Jun 2023

PONE-D-23-14306Comparison of five different Disseminated Intravascular Coagulation Criteria in predicting mortality in Patients with SepsisPLOS ONE

Dear Dr. Bin Zafar Mahmood,

Thank you for submitting your manuscript to PLOS ONE. After careful consideration, we feel that it has merit but does not fully meet PLOS ONE’s publication criteria as it currently stands. Therefore, we invite you to submit a revised version of the manuscript that addresses the points raised during the review process.

We look forward to receiving your revised manuscript.

Kind regards,

Yoshihisa Tsuji

Academic Editor

PLOS ONE

https://journals.pl2.os.org/plosone/s/file?id=wjVg/PLOSOne_formatting_sample_main_body.pdf and

Additional Editor Comments:

There are several problems in methodology, and it does not reach a publishable level. The points raised by the reviewers are essential; please respond to these questions.

Reviewers' comments:

Reviewer's Responses to Questions

**Comments to the Author**

1. Is the manuscript technically sound, and do the data support the conclusions?

Reviewer #1: No

Reviewer #2: Partly

2. Has the statistical analysis been performed appropriately and rigorously? 

Reviewer #1: Yes

Reviewer #2: No

3. Have the authors made all data underlying the findings in their manuscript fully available?

Reviewer #1: No

Reviewer #2: Yes

4. Is the manuscript presented in an intelligible fashion and written in standard English?

Reviewer #1: Yes

Reviewer #2: Yes

5. Review Comments to the Author

Reviewer #1: The Manuscript compared the performance of five DIC diagnostic criteria focusing on the prediction of mortality in Pakistan. A retrospective cohort study using electronic medical records was conducted. The authors concluded that Japanese Association for Acute Medicine (JAAM) and Korean Society on Thrombosis and Hemostasis (KSTH) criteria were most valuable diagnostic criteria for screening septic patients for DIC in Pakistan population.

Major Comments

1) The details what is sepsis in ICD-9 was not written specifically. A description of what sepsis is included is needed.

2) There is no data description regarding the implementation of DIC score assessment. Basically, DIC scores do not use the parameters from time points apart in time. Are the parameters measured at the same time? What percentage of the evaluation was done on day 1？

3) A significant confounding factor regarding DIC diagnosis and mortality is intervention for DIC. The information about ICU admission rates and treatment for DIC needed.

4) It is not mentioned whether the DIC score evaluator was aware of the information regarding patient outcomes.

5) All of 5 criteria seem to have low predictive accuracy. Nevertheless, the authors recommend the use of the DIC score, but what is the advantage of recommending its use over other mortality predictors? I think the conclusion should be changed.

Minor Comments

1) For external validation, information on the clinical setting is needed. For example, information on patient transport routes and ICU admission rates is needed.

2) No unit description for lactate in Line 168.

Reviewer #2: This study compared the 5 different DIC criteria, and revealed the diagnostic accuracy for septic DIC as well as prediction of mortality. Unfortunately, I cannot accept this study for the following some reasons. Please see comments below.

＜Major revision＞

1) Internal validity

In this study, the relationship between DIC and patient outcomes were analyzed, but never analyzed some important confounding factors such as comorbidities.

2) External validity

It is difficult to apply the results of the 222 patients included in the study to patients with sepsis in general. The number included in the study is too small to generalize the result.

3) Feasibility

The authors recommend the combined use of JAAM and KSTH criteria for septic DIC. This means “ double standard “ in the diagnosis, which is inconsistent with current practice of sepsis.

4) Novelty

The authors mentioned Reference No.8 evaluating 5 DIC criteria like them, and the results were different between them. They never explained the reason of the inconsistent results. The novelty was unclear.

6. PLOS authors have the option to publish the peer review history of their article (what does this mean?). If published, this will include your full peer review and any attached files.

Reviewer #1: No

Reviewer #2: No

---

## [Author Response · Author response to Decision Letter 0]

12 Aug 2023

1 The details of what sepsis is in ICD-9 was not written specifically. A description of what sepsis is included is needed. 

We acknowledge this shortcoming. Dombrovskiy definition of sepsis was taken and has now been explained in methodology. The new reference has also been included (#11) Page 6 Line 111-115

2 There is no data description regarding the implementation of DIC score assessment. Basically, DIC scores do not use the parameters from time points apart in time. Are the parameters measured at the same time? What percentage of the evaluation was done on day 1？ 

We acknowledge that this data is missing. This has now been included “All the elements used to assess DIC criteria for each patient originated from a single day. On the initial day, 44.3% underwent DIC evaluation based on clinical judgment, while on the second day, the assessment was conducted for 33.3%. On the third day, 13.1% underwent evaluation, followed by 4.2% on the fourth day, and 3.4% on the fifth day. The remaining four patients underwent DIC evaluation on days 6, 8, 10, and 11, respectively.” Page 7, Line 139-143

3 A significant confounding factor regarding DIC diagnosis and mortality is intervention for DIC. The information about ICU admission rates and treatment for DIC needed. 

We acknowledge this deficiency. The data for ICU admission rates and outcome was available and has been included. 89 patients (40.1%) were intubated and experienced an Intensive Care Unit stay. However, treatment received has not been included and has been acknowledged as a limitation. Page 11, Line 184 and Page 19, Line 318-321

4 It is not mentioned whether the DIC score evaluator was aware of the information regarding patient outcomes. 

Since this is a retrospective study, medical record files were systematically reviewed and patients meeting inclusion criteria were included. Scores were applied as per the criteria and outcome was noted from the records by the same data collector. This detail has been included in the methodology also. Page 6, Line 124-127

5 All of 5 criteria seem to have low predictive accuracy. Nevertheless, the authors recommend the use of the DIC score, but what is the advantage of recommending its use over other mortality predictors? I think the conclusion should be changed. 

We have changed the conclusion to state "Our study findings indicate that among the five evaluated models of DIC, KSTH demonstrated superior sensitivity and accuracy, while R-JAAM exhibited the highest specificity and positive predictive value for mortality prediction in sepsis patients as compared to others. Our study results highlight the need for large scale studies to validate these findings before generalizing it and making it a part of local guidelines." Page 20, Line 336-341

6 For external validation, information on the clinical setting is needed. For example, information on patient transport routes and ICU admission rates is needed. 

We have added more information regarding the clinical setting. "Aga Khan University Hospital is one of the largest academic tertiary care centers in South Asia. It is a 600 bed facility with a state of the art emergency department and rooms of different acuities; low-monitoring ward beds with a nurse ratio of approximately 5:1, high dependency units with 24 hour cardiac monitoring, non-invasive mechanical ventilation facilities and nurse ratio of 5:2, and intensive care units (ICU) with mechanical ventilator facilities and nurse ratio of 1:1. In addition, the hospital has a rapid response team consisting of an ICU nurse and doctor and is the first responder to hospital areas in case of any emergency. The hospital also has a robust transport mechanism for patients who need to be shifted to ICU. Such patients are accompanied by an Advanced cardiac life support (ACLS) certified nurse and doctor along with the RRT nurse." Page 5, Line 100-109

7 No unit description for lactate in Line 168. 

The unit for lactate (mM/L) has been added Page 11, Line 189-190

1 Internal validity

In this study, the relationship between DIC and patient outcomes were analyzed, but never analyzed some important confounding factors such as comorbidities. 

We agree that data on comorbidities is necessary. The two most important comorbidities diabetes and hypertension have now been included in Table 2 Page 12, Line 213

2 External validity

It is difficult to apply the results of the 222 patients included in the study to patients with sepsis in general. The number included in the study is too small to generalize the result. 

We have changed the conclusion to state

Our study findings indicate that among the five evaluated models of DIC, KSTH demonstrated superior sensitivity and accuracy, while R-JAAM exhibited the highest specificity and positive predictive value for mortality prediction in sepsis patients as compared to others. Our study results highlight the need for large scale studies to validate these findings before generalizing it and making it a part of local guidelines. Page 20, Line 336-341

3 Feasibility

The authors recommend the combined use of JAAM and KSTH criteria for septic DIC. This means “ double standard “ in the diagnosis, which is inconsistent with current practice of sepsis. 

We have changed the conclusion to state

Our study findings indicate that among the five evaluated models of DIC, KSTH demonstrated superior sensitivity and accuracy, while R-JAAM exhibited the highest specificity and positive predictive value for mortality prediction in sepsis patients as compared to others. Our study results highlight the need for large scale studies to validate these findings before generalizing it and making it a part of local guidelines. Page 20, Line 336-341

4 Novelty

The authors mentioned Reference No.8 evaluating 5 DIC criteria like them, and the results were different between them. They never explained the reason of the inconsistent results. The novelty was unclear. 

We have explained the reason with a source reference. "Overall, comparison between two studies would be difficult as one was prospective while the other was retrospective. Moreover, it is important to note that each population has different racial characteristics and hence different thrombolytic mechanisms." Page 17, Line 264-267

---

## [Decision Letter · Decision Letter 1]

4 Sep 2023

PONE-D-23-14306R1Comparison of five different Disseminated Intravascular Coagulation Criteria in predicting mortality in Patients with SepsisPLOS ONE

Dear Dr. Mahmood,

Thank you for submitting your manuscript to PLOS ONE. After careful consideration, we feel that it has merit but does not fully meet PLOS ONE’s publication criteria as it currently stands. Therefore, we invite you to submit a revised version of the manuscript that addresses the points raised during the review process.

Reviewer ＃１ pointed out important points below. Please check and respond again.

We look forward to receiving your revised manuscript.

Kind regards,

Yoshihisa Tsuji

Academic Editor

PLOS ONE

Reviewers' comments:

Reviewer's Responses to Questions

**Comments to the Author**

1. If the authors have adequately addressed your comments raised in a previous round of review and you feel that this manuscript is now acceptable for publication, you may indicate that here to bypass the “Comments to the Author” section, enter your conflict of interest statement in the “Confidential to Editor” section, and submit your "Accept" recommendation.

Reviewer #1: All comments have been addressed

Reviewer #2: All comments have been addressed

2. Is the manuscript technically sound, and do the data support the conclusions?

Reviewer #1: No

Reviewer #2: Yes

3. Has the statistical analysis been performed appropriately and rigorously? 

Reviewer #1: No

Reviewer #2: Yes

4. Have the authors made all data underlying the findings in their manuscript fully available?

Reviewer #1: Yes

Reviewer #2: Yes

5. Is the manuscript presented in an intelligible fashion and written in standard English?

Reviewer #1: Yes

Reviewer #2: Yes

6. Review Comments to the Author

Reviewer #1: Major Comments

1) Thank you for providing more information on the ICD-9 criteria for inclusion based on Dombrovskiy's work, but reading Dombrovskiy's article, it seems to cover severe sepsis based on sepsis-1. Is it correct to understand that this corresponds to "sepsis" according to today's sepsis-3 definition? A definition of sepsis is needed.

2) According to the #11 literature, the classification is by ICD-9 CM Code, not ICD-9.

Which coding is correct?

3) Thank you for describing the data on the implementation of the DIC score assessment. I understand that the evaluations are carried out at different points in time, does this mean that the worst values are used as data? Do you mean the initial assessment? Please describe it in a way that the reader can understand.

4）Regarding results. I calculated the performance of the JMHW from the data in the manuscript, but there is concern that the results for specificity, positive predictive value and accuracy are incorrect. My calculations yielded a specificity of 61.8%, a PPV of 78.0% and an accuracy of 60.3%.

If the results of the calculations are in error, the performance difference in the five criteria appears to be narrowing.

If the results change, the conclusions should be re-considered.

5）It should be stated in the Limitation that the timing of coagulation testing is dependent on clinician bias and that the therapeutic interventions for the diagnosis of DIC are unknown.

Reviewer #2: The authors answered all questions adequately and reflected them in the text. I have no further comments.

7. PLOS authors have the option to publish the peer review history of their article (what does this mean?). If published, this will include your full peer review and any attached files.

Reviewer #1: No

Reviewer #2: No

---

## [Author Response · Author response to Decision Letter 1]

19 Oct 2023

Reviewer 1: 

1 Thank you for providing more information on the ICD-9 criteria for inclusion based on Dombrovskiy's work, but reading Dombrovskiy's article, it seems to cover severe sepsis based on sepsis-1. Is it correct to understand that this corresponds to "sepsis" according to today's sepsis-3 definition? A definition of sepsis is needed. 

Thank you for taking the time to review our article, and we appreciate your feedback. Your attention to detail is invaluable in helping us improve the clarity of our work.

As required the definition of sepsis has been added as mentioned below with the appropriate reference. 

“Sepsis was defined as life-threatening organ dysfunction caused by a dysregulated host response to infection”

However, global literature itself acknowledges the inconsistent strategies in selecting ICD codes for sepsis and the resulting problem. Most of these ICD codes for sepsis have been categorized before 2016 when Sepsis-3 definition was not in use. 

The first JAMA paper (Singer M, et al. The Third International Consensus Definitions for Sepsis and Septic Shock (Sepsis-3). JAMA. 2016 Feb 23;315(8):801-10.) on Sepsis-3 defined sepsis as per the definition mentioned above and advised to use only two explicit codes (995.92 and 785.52) for sepsis categorization. 

However, a most recent article by Rudd and colleagues (Rudd et al. Global, regional, and national sepsis incidence and mortality, 1990–2017: analysis for the Global Burden of Disease Study. Lancet. 2020; 395: 200-211) and an extended commentary by Duke and colleagues accepts that using only explicit codes mentioned above may be inconsistent, unreliable and underestimate the true prevalence of sepsis. 

The advice by these authors is that sepsis diagnosis should require two simultaneous diagnosis codes: one for an infectious disease and another for an acute organ dysfunction diagnosis. Unfortunately, the new categorization being done now is based on ICD-10-CM codes, whereas our hospital has switched to ICD-10-CM codes from 2022 and all previous coding has been done based on ICD-9-CM codes. Therefore, we utilized the existing work of Dombrovskiy's for ICD-9-CM code list from the work of Rhee and others, for the Centers for Disease Control and Prevention Epicenters Program as mentioned in the manuscript. This ICD-9-CM coding was also approved and provided by our Hospital Management Information System (HIMS) office tasked with maintaining all the patient data and categorizing them according to ICD codes.

I would like to clarify that the ICD-9-CM codes in this article mentioned under the heading of Dombrovskiy's definition do enroll severe sepsis patients but based on the same definition of having a source of infection along with an evidence of organ dysfunction. This definition is comparable to sepsis definition as per sepsis-3 which is applicable nowadays as also seen in the JAMA paper quoted above.

However, to enhance the validity of our study, we also included the modified Martin Criteria as done by Bouza and colleagues, Fortunately, there were only a few additional ICD-9-CM codes that were added as mentioned in text

“and Modified Martin Criteria (Infection codes: 038.0 - 038.4, 038.8, 038.9, 003.1, 020.2 , 036.2, 036.3, 054.5, 098.89, 112.5, 112.81, 117.9, 790.7, 995.91 AND organ dysfunction codes: 518.81, 518.82, 518.84, 518.85, 786.09, 799.1, 785.5 with all sub codes, 458 , 796.3 , 584 with all sub codes, 580 , 570, 572.2, 573.3, 286.6, 286.9, 287.3-5, 293, 348.1, 348.3, 357.82, 780.01, 780.09, 276.2)”

However, importantly no additional entries were obtained in the study cohort likely because the coding done by the HIMS office is based on the Dombrovskiy's code list. 

To further add, we have added the ICD coding issues as a limitation in our manuscript

“Moreover, ICD-9-CM coding for sepsis has been found to be categorized differently across literature and needs to be looked thoroughly between different institutions.”

I believe that this explanation would help in clarifying any remaining issues regarding the sepsis definition in our manuscript. Moreover, the study’s main objective is to compare the different DIC criteria, and we believe that the study methodology can easily be replicated at another institution using ICD-9-CM or ICD-10-CM codes as per the hospital ICD code list. 

Page 6, Line 109-121 and Page 20, Line 321-322

2 According to the #11 literature, the classification is by ICD-9 CM Code, not ICD-9.

Which coding is correct? 

We apologize for this misconception.

The ICD-9 is just an abbreviated form of the full ICD-9-CM Code. ICD-9-CM is the official system of assigning codes to diagnoses and procedures associated with hospital utilization. This is abbreviated to ICD-9.

To make it easy and clear for the readers we have used the term “ICD-9-CM” all over the manuscript now. 

3 Thank you for describing the data on the implementation of the DIC score assessment. I understand that the evaluations are carried out at different points in time, does this mean that the worst values are used as data? Do you mean the initial assessment? Please describe it in a way that the reader can understand. 

We appreciate your diligence in seeking clarification. 

The following text is already included in the methodology section but has been highlighted again in the manuscript.

“All the elements used to assess DIC criteria for each patient originated from a single day. On the initial day, 44.3% underwent DIC evaluation based on clinical judgment, while on the second day, the assessment was conducted for 33.3%. On the third day, 13.1% underwent evaluation, followed by 4.2% on the fourth day, and 3.4% on the fifth day. The remaining four patients underwent DIC evaluation on days 6, 8, 10, and 11, respectively. Hence, these investigations were carried out on the day when the clinician suspected the patient might be experiencing DIC. These values are not the worst reported values during the admission but were done on a single day” 

Page 7, Line 144-151

4 Regarding results. I calculated the performance of the JMHW from the data in the manuscript, but there is concern that the results for specificity, positive predictive value and accuracy are incorrect. My calculations yielded a specificity of 61.8%, a PPV of 78.0% and an accuracy of 60.3%.

If the results of the calculations are in error, the performance difference in the five criteria appears to be narrowing.

If the results change, the conclusions should be re-considered. 

We are extremely thankful for highlighting the numerical inaccuracies within Table 4. These errors have been thoroughly reviewed and rectified in the manuscript. We can confirm that all other numerical data has been meticulously scrutinized and found to be accurate. 

Fortunately, these corrections do not alter the overall conclusions of the manuscript, as our study does not offer a definitive criterion based on our findings, rather encourage the researchers to consider replicating this study in a prospective manner to yield more robust results considering the high overall mortality. However, the required value changes have been made in the table and text.

5 It should be stated in the Limitation that the timing of coagulation testing is dependent on clinician bias and that the therapeutic interventions for the diagnosis of DIC are unknown. 

We do agree with your opinion on both accounts and have made the changes. This was already mentioned in the manuscript, but we have further clarified it based on your valuable suggestion. 

The following text has been added in limitations to further clarify the limitations.

“The selection of patients who underwent coagulation testing relied on the clinical judgment of physicians, representing a notable constraint in our study.”

Regarding therapeutic interventions the following lines were already present. 

“Moreover, our study also did not include the treatment regimens for these patients due to the variability and since there is yet to be an international consensus on whether DIC should be a therapeutic target with anticoagulant therapy” 

Page 20, Line 324-330

Reviewer 2: The authors answered all questions adequately and reflected them in the text. I have no further comments.

Thank you for taking the time to review our article, and we appreciate your feedback.

---

## [Decision Letter · Decision Letter 2]

15 Nov 2023

Comparison of five different Disseminated Intravascular Coagulation Criteria in predicting mortality in Patients with Sepsis

PONE-D-23-14306R2

Dear Dr. Mahmood,

We’re pleased to inform you that your manuscript has been judged scientifically suitable for publication and will be formally accepted for publication once it meets all outstanding technical requirements.

Kind regards,

Yoshihisa Tsuji

Academic Editor

PLOS ONE

---

## [Editor Report · Acceptance letter]

20 Nov 2023

PONE-D-23-14306R2 

Comparison of five different Disseminated Intravascular Coagulation Criteria in predicting mortality in Patients with Sepsis 

Dear Dr. Mahmood:

I'm pleased to inform you that your manuscript has been deemed suitable for publication in PLOS ONE. Congratulations! Your manuscript is now with our production department. 

Kind regards, 

on behalf of

Professor Yoshihisa Tsuji 

Academic Editor

PLOS ONE